# Daily actual evapotranspiration estimation of different land use types based on SEBAL model in the agro-pastoral ecotone of northwest China

**Liangyan Yang** [1,2]\*, **Jianfeng Li**[1,2], **Zenghui Sun**[1,2]\*, **Jinbao Liu**[1,3], **Yuanyuan Yang**[4]◉, **Tong Li**[4]◉

**1** Shaanxi Provincial Land Engineering Construction Group Co., Ltd, Xi'an, Shaanxi, China, **2** Institute of Land Engineering and Technology, Shaanxi Provincial Land Engineering Construction Group Co., Ltd., Xi'an, Shaanxi, China, **3** Xi'an University of Technology, Xi'an, Shaanxi, China, **4** School of Earth Science and Resources, Chang'an University, Xi'an, Shaanxi, China

◉ These authors contributed equally to this work.
\* 2016127008@chd.edu.cn (LY); sunzenghui061@126.com (ZS)

**Data Availability Statement:** All relevant data are within the paper and its Supporting information files.

## Abstract

Evapotranspiration (ET) plays a crucial role in hydrological and energy cycles, as well as in the assessments of water resources and irrigation demands. On a regional scale, particularly in the agro-pastoral ecotone, clarification of the distribution of surface ET and its influencing factors is critical for the rational use of water resources, restoration of the ecological environment, and protection of ecological water sources. The SEBAL model was used to invert the regional ET based on Landsat8 images in the agro-pastoral ecotone of northwest China. The results were indirectly verified by monitoring data from meteorological stations. The correlation between ET and surface parameters was analyzed. Thus, the main factors that affect the surface ET were identified. The results show that the SEBAL model determines an accurate inversion, with a correlation coefficient of 0.81 and an average root mean square error of 0.9 mm/d, which is highly suitable for research on water resources. The correlation coefficients of normalized vegetation index, surface temperature, land surface albedo, net radiation flux with daily ET were 0.5830, 0.8425, 0.3428 and 0.9111, respectively. The normalized vegetation index and the net radiation flux positively correlated with the daily ET, while the surface temperature and land surface albedo negatively correlated with the daily ET. The correlation from strong to weak is the net radiation flux > surface temperature > normalized vegetation index > surface albedo. In terms of spatial distribution, the daily ET of water was the highest, followed by woodland, wetland, cropland, built-up land, shrub land, grassland and bare land. However, the SEBAL model overestimates the inversion of daily ET of built-up land.

**Funding:** This paper was supported by the Fund for Less Developed Regions of the National Natural Science Foundation of China (No. 42167039), Technology Innovation Center for Land Engineering and Human Settlements, Shaanxi Land Engineering Construction Group Co., Ltd and Xian Jiaotong University (2021WHZ0088, 2021WHZ0091) and Scientific Research Item of Shaanxi Provincial Land Engineering Built-up Group (DJNY2021-33). The funder provided support in the form of salaries for authors [Yang LY, Li JF, Sun ZH, Liu JB], but did not have any additional role in the study design, data collection and analysis, decision to publish, or preparation of the manuscript. The specific roles of these authors are articulated in the 'author contributions' section.

**Competing interests:** Institute of Land Engineering and Technology, Shaanxi Provincial Land Engineering Construction Group Co., Ltd and Shaanxi Provincial Land Engineering Construction Group Co., Ltd do not alter our adherence to PLOS ONE policies on sharing data and materials. The authors have declared that no competing interests exist.

## Introduction

Evapotranspiration (ET) is an important part of the global water cycle [1]. It is the channel that transforms surface and atmospheric water, which directly affects the spatial distribution of global precipitation and vegetation [2,3]. Studies have shown that > 60% of the precipitation that reaches the surface through condensation returns to the atmosphere through the process of ET, and this proportion can be as high as 90% owing to a reduction in precipitation and a higher ET in arid areas [4–6]. Solar short-wave radiation energy returns to the atmosphere in the form of latent heat during the process of ET on the surface, thus, forming an energy cycle process. Therefore, the surface ET plays an important role in the global water and energy cycles [7]. Simultaneously, the actual surface ET is a key parameter for research on environmental changes [8], hydrological processes [9], drought trends [10,11], and the potential evaluation of agricultural development [12,13]. Therefore, the study of surface ET is highly important scientifically to ensure the rational use of water resources and the protection of ecosystems in the study area [14,15].

Owing to the complexity of the surface, current ET observation methods include lysimeters [16], eddy correlator large aperture scintillation flux meters [17] and other methods. Such ET monitoring methods can obtain the ET more accurately at a specific location, but simultaneously, it is time-consuming, labor-intensive and expensive, and only a single point of ET data can be obtained. Therefore, the acquisition of regional surface ET data that is highly accurate over a continuous time-scale is always the key to solving difficult issues of research in the fields of meteorological science, hydrology, agriculture and geography. The development of remote sensing technology provides a new platform and way to obtain the actual surface ET, and it solves the problem of a lack of measured data owing to geographical reasons. In recent years, researchers have made substantial progress in utilizing remote sensing ET models for research, which successively established the surface energy balance index (SEBI) model [18], simplified surface energy balance index (S-SEBI) model [19], surface energy balance algorithm for land (SEBAL) model [20] and surface energy balance system (SEBS) model [21]. Among them, the SEBAL model benefits from a clearer mechanism, fewer model input parameters, simple data acquisition, high inversion accuracy, and a wide application range. It has become one of the most commonly used remote sensing methods to determine the inversion of surface ET [22].

However, the accuracy of SEBAL model varies in the estimation of ET for different products and environmental conditions. Liu and Hu investigated the effects of land use/cover change and climate change on wetland ET using SEBAL, and the results showed that the average relative error of regional ET estimated by the SEBAL model was 9.01% [23]. Rahimzadegan et al. studied the efficiency of SEBAL algorithm in estimating the ET of pistachio crops. The results of this study indicated that the determination coefficient and Root Mean Square Error of the estimated actual ET for pistachio plants was 0.8 and 2.5 mm, respectively [24]. The agro-pastoral ecotone of northwest China receives little rainfall, which is distributed unevenly, and has a strong surface ET and an extremely fragile ecological environment. A shortage of water is the most serious problem in this area. In addition, the research on the influence factors of ET mostly focuses on meteorological factors [25,26], and there is a lack of in-depth research on the influence of surface parameters on ET.

The objective of this study was to explore the applicability of SEBAL model and analyze the influencing factors in a wind and sandy grass beach area of northwest China. Therefore, in this study, (1) Landsat8 imagery and meteorological data were used as the source of data to estimate daily ET based on the SEBAL model; (2) the accuracy of estimation of daily ET was verified and used to analyze its spatiotemporal distribution, and (3) the influence of surface

parameters on the daily ET was studied, providing an analysis of the response of daily ET to different land use types. These conclusions will enhance our understanding of the factors that affect ET and thus, provide a basis for the sustainable development of water resources and the restoration of the ecological environment in arid and semiarid regions in northwest China.

## Materials and methods

### Study area

The study area (107˚58′-109˚15′E, 37˚31′-38˚49′) is located at the junction of Yulin City and Ordos City, China (Fig 1) in a sandy and grassy beach area southeast of Mu Us Sandy Land. It has a typical farming-pastoral transition zone with a composite underlying surface of bare land-grassland-farmland in China. The study area has a typical continental monsoon climate. The northwest is arid and lacks water, which results in infrequent precipitation. Consequently, the ecological environment is fragile. The land use types are primarily grassland and bare land. The southeast is low-lying, with rivers that pass through, and the land use types are primarily grassland and cropland. The average annual temperature is approximately 7˚C and the average annual precipitation is between 350–450 mm, which decreases from southeast to northwest in the study area. The interannual precipitation is highly uneven with a large seasonal variation. The rainy season is from July to September, and it comprises more than 50% of the rain that falls during a whole year. The potential ET is relatively large, and the annual average ET is 2,000–3,000 mm in the study area. The sparse precipitation and strong ET activities are the main reasons for the drought in study area.

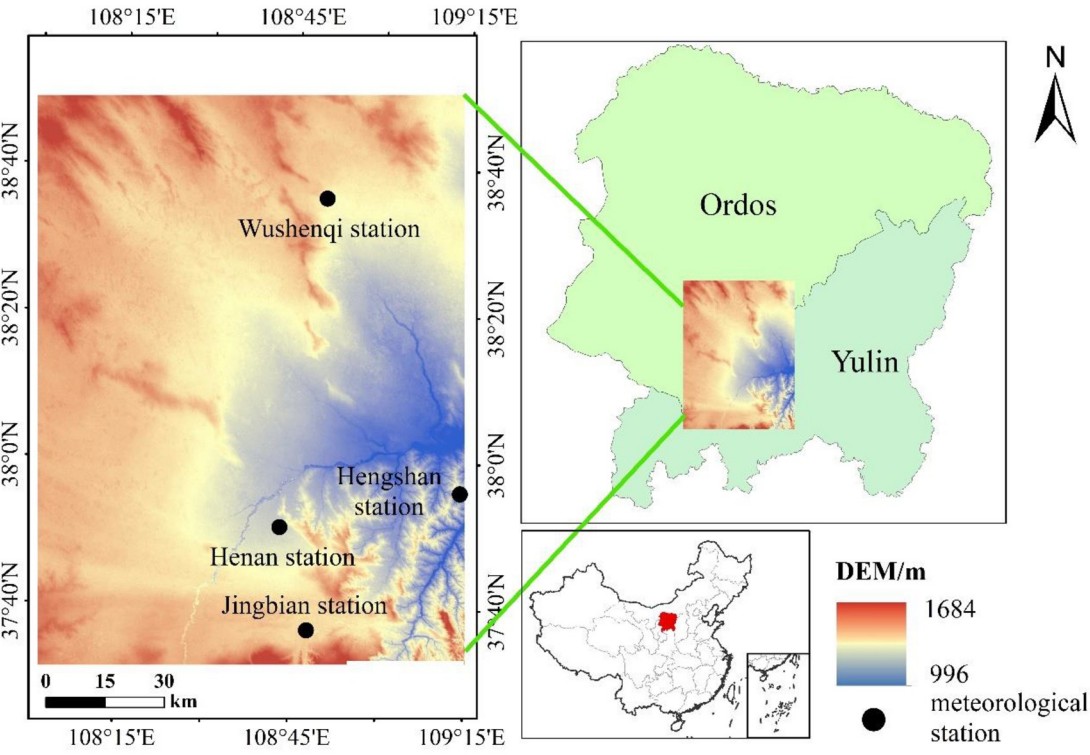

**Fig 1. Location of the study area in agro-pastoral ecotone of northwest China.**

## Data sources

In this study, the agro-pastoral ecotone of northwest China was used as the study area, and the Landsat8 images that cover the study area were numbered 128,033 and 128,034. They are all freely available at the Geospatial Data Cloud (http://www.gscloud.cn). Since the data is affected by cloud coverage, and parts of it are missing, it is impossible to obtain the image data for each period of the same year. To ensure that the data images can cover the study area, this study screened out 2017-1-2, 2016-4-21, 2017-5-26, 2017-7-13, 2016-9-28, 2016-12-17 and six other Landsat8 original remote sensing images. The selected data covered the four seasons of the year and are suitable for the study of the spatiotemporal distribution of surface ET. The Landsat8 remote sensing data requires preprocessing, such as radiometric calibration, atmospheric correction and image mosaics, before it can be substituted into the model. ENVI software (L3Harris Geospatial, Boulder, CA, USA) was used to process the images in this research.

## Principle of the SEBAL model

In 1995, Bastiaanssen proposed the SEBAL inversion model, which was verified and optimized in 1998. In recent years, the surface ET model has been effectively developed and improved. The SEBAL model has been utilized for its advantages and widely used throughout the world. It has produced good inversion results and has become a high-precision regional ET remote sensing inversion model. The research foundation of the SEBAL model is the energy balance equation, which is composed of four parts, including net radiance, soil heat, sensible heat and latent heat fluxes, which form a closed-loop energy transmission. The formula is shown as follows:

$$R_n = H + G_n + \lambda ET \tag{1}$$

Where $R_n$ is the net solar radiation flux (W/m$^2$); $G_n$ is the soil heat flux (W/m$^2$); $H$ is the turbulent sensible heat flux (W/m$^2$); $\lambda ET$ is the turbulent latent heat flux (W/m$^2$), and $\lambda$ is the coefficient of latent heat of vaporization of water, which is a function of temperature.

   **Net radiant flux ($R_n$).**   The net surface radiant flux is expressed per unit area, which includes the difference between total energy received from the solar short-wave radiation and atmospheric long-wave radiation on the surface and the long-wave radiation emitted by the surface, reflected solar short-wave radiation and atmospheric long-wave radiation [27]. The process of calculation is as follows:

$$R_n = (1 - \alpha) \times R_{swd\downarrow} + R_{lwd\downarrow} - R_{lwd\uparrow} - (1 - \varepsilon) \times R_{lwd\downarrow} \tag{2}$$

$$R_{swd\downarrow} = G_{sc} \times cos\theta \times d \times \tau_{sw} \tag{3}$$

$$d = 1 + 0.033cos(DAY \times 2\pi/365) \tag{4}$$

$$R_{lwd\downarrow} = \varepsilon_a \times \sigma \times T_a^4 \tag{5}$$

$$\varepsilon_a = 9.2 \times 10^\wedge[-6 \times (T_a - 273.15)^2] \tag{6}$$

$$R_{lwd\uparrow} = \varepsilon \times \sigma \times LST^4 \tag{7}$$

Where $R_{swd\downarrow}$ is the short-wave radiation energy that the sun reaches the surface (W/m$^2$); $R_{lwd\downarrow}$ is the long-wave radiation from the atmosphere to the surface (W/m$^2$); $R_{lwd\uparrow}$ is the ground upgoing long-wave radiation (W/m$^2$); $\alpha$ is the surface albedo; $\varepsilon$ is the surface emissivity; $G_{sc}$ is

the solar constant, with a value of 1,367 W/m$^2$; θ is the zenith angle of sun; d is the distance factor between the sun and the earth; $\tau_{sw}$ is the atmospheric transmittance; DAY is the number of days of the image in the current year; $\varepsilon_a$ is the specific emissivity of the atmosphere; σ is a constant with a value of 5.67×10$^{-8}$ W/(m$^2$·K$^4$); $T_a$ represents the temperature at the ground reference altitude, and LST represents the surface temperature.

**Soil heat flux (G$_n$).**   The soil heat flux is the difference between surface and soil temperatures that enables the transmission of energy. A small part of the energy lost in the surface soil during this process is primarily affected by topography and land use types. The soil heat flux generally occupies a relatively small proportion in the energy balance equation. Factoring in the greater difficulty in its direct calculation, Bastiaanssen used calculations of net radiant flux, vegetation index, land surface temperature and land surface reflectance in the SEBAL model [28], and the calculations are shown as follows:

$$G_n = R_n \times \frac{T_s - 273.15}{\alpha}(0.0038\alpha + 0.0074\alpha^2)(1 - 0.98 \times NDVI^4) \tag{8}$$

Where NDVI is the normalized vegetation index.

**Sensible heat flux (H).**   Sensible heat flux is the temperature difference between the air and surface temperatures, and the energy that flows into the air from the surface is in the form of vertical transmission. The sensible heat flux is primarily related to the ground temperature difference, density, aerodynamic impedance and other parameters, and it is calculated is as follows:

$$H = \frac{\rho_{air} \times C_P \times d_T}{r_{ah}} \tag{9}$$

Where $C_P$ is the specific heat capacity of the atmosphere; $\rho_{air}$ is the density of the atmosphere; $r_{ah}$ is the aerodynamic impedance (s·m$^{-1}$), and $d_T$ is the ground temperature difference (K). $r_{ah}$ and $d_T$ are unknown parameters. The specific process for calculation has been previously described [29].

**Latent heat flux (λET).**   Latent heat flux is the heat exchange per unit area at constant temperature, which is the most important parameter in the inversion study of surface ET. It is the energy absorbed during the vaporization of water. Based on the surface energy balance formula in the remote sensing inversion ET model, the latent heat flux is obtained as follows:

$$\lambda ET = R_n - H - G_n \tag{10}$$

$\lambda ET$ represents the instantaneous latent heat flux (W/m$^2$).

**Daily ET.**   The surface radiation flux is substantially affected by factors, such as surface temperature, wind speed, and surface coverage, and cannot be obtained directly. The difference in ratio of latent heat flux to the net radiant and soil heat fluxes in a day always remain basically stable, i.e., the theory of constant evaporation ratio and the instantaneous ET can be calculated by the theory of constant evaporation ratio, which can be extended to the daily ET [30,31]. The formula is calculated as follows:

$$EF = \frac{R_n - G_n - H}{R_n - G_n} = \frac{\lambda ET}{R_n - G_n} = EF_{24} \tag{11}$$

$$ET_{24} = \frac{86400 \times EF_{24}}{\lambda} \times (R_{n24} - G_{n24}) \tag{12}$$

Where $EF$ represents the instantaneous evaporation ratio; $EF_{24}$ represents the entire day evaporation ratio; $R_{n24}$ represents the daily net radiant flux (W/m$^2$), and $G_{n24}$ represents the daily soil heat flux (W/m$^2$).

## Results

### Verification of the accuracy of daily ET

The verification and evaluation of the inversion accuracy of the surface ET model is the focus and difficulty of the research on ET. To verify the reliability of the daily ET retrieved by the SEBAL model, the water surface evaporation monitored by four weather stations in the study area was selected for relative verification. However, the evaporation on the water surface that was measured was significantly higher than the actual evaporation on the surface, and the water surface evaporator of the weather station has two types that include E-601 and 20cm-caliber. There are differences in the installation plan, structure and observational methods of the two instruments. There are large deviations in the verification of data when the data of weather stations is used directly. Therefore, to more accurately explore the inversion accuracy of the SEBAL model and maintain the continuity and consistency of the data at each station, it is necessary to explore the conversion coefficients of the two surface evaporators and unify the data. E-601 and 20cm-caliber meteorological stations, such as those at the Hengshan, Jingbian, Shenmu, Dingbian, Yanchi and Etuoke Banner Stations in China were compared in the study and surrounding areas. The monitoring data were analyzed and indicated that the average conversion coefficient of the two surface evaporators was 0.63, and the determination coefficient $R^2$ was 0.95, indicating that the conversion coefficients of the E-601 type and the 20cm-caliber small evaporator are highly reliable, respectively, and can meet the requirements of the two types of surface evaporators.

The monitoring data of 20cm-caliber small evaporator was converted into the E-601 monitoring data using the conversion coefficients of the two surface evaporators. This enabled the use of data to verify the SEBAL model inversion result. Since the meteorological sites were small meteorological stations that were established by research teams in Wushenqi and Henan in China, they did not have evaporation dishes to obtain the data on water surface evaporation. The only data involved in the validation were the monitoring data from the meteorological stations in Hengshan and Jingbian, China. Fig 2 is a scatter diagram of the water surface evaporation of the meteorological station and the inversion results. This figure indicates that the actual surface ET based on the SEBAL model is highly correlated with the surface water surface ET. The coefficient of determination was 0.81, and the average root mean square error was 0.9 mm/d. These values indicate that the SEBAL model can be used to retrieve the surface ET in agro-pastoral ecotone of northwest China with a high degree of accuracy.

### Spatiotemporal distribution of the daily ET

In this study, the daily ET of the study area was obtained based on the theory of constant ET ratio. Table 1 shows the data on daily ET in different periods in the study area. An examination of the distribution revealed a pattern in the daily ET in the northwest agro-pastoral ecotone area. The pattern included July (5.17 mm/d)＞May (3.51 mm/d)＞September (3.31 mm/d)＞ April (2.02 mm/d)＞December (0.72 mm/d)＞January (0.56 mm/d). The possible reason is related to the local climate change during the four seasons. The temperature and precipitation were the lowest during the winter in January and December, and insufficient hydrothermal conditions limited the ET process. In April and May, the air temperature began to increase; the vegetation gradually turned green, and the crops were just beginning to be cultivated and sprouting. These factors indicate that the ET gradually increased. Since July is in the summer in this study area, the monthly average temperature reaches the highest value of the year, and this is the rainy season. The vegetation grows vigorously, and there is sufficient soil moisture. The hydrothermal conditions were the highest of the year. Therefore, the study area had its

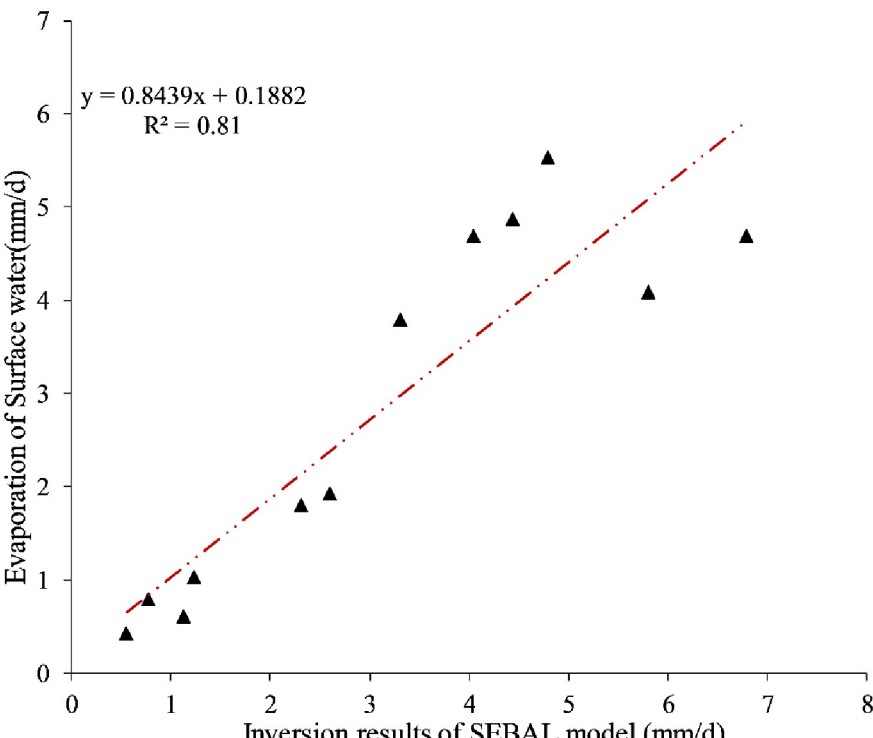

**Fig 2. Validation scatter plot of the SEBAL model inversion results and meteorological data.**

highest daily ET in the summer. In September, the vegetation in the sand of study area began to senesce; the temperature decreased; the rainfall was significantly reduced; the hydrothermal conditions were all lower than those in the summer, and the daily ET was less than in the summer. In summary, hydrothermal conditions are the main factor that affect the changes in evapotranspiration during the year.

Fig 3 shows the spatial distribution of the daily ET in different periods in the study area. The evapotranspiration varied with regions within the study area. It was higher overall in the southeast and lower in the northwest, and such a pattern of distribution corresponds to the terrain of the study area. There are rivers that pass through the southeast, and there are many areas of crops, woodland and grassland. The type of land use in northwest China primarily included denuded dunes or bare land. Thus, there is less vegetation coverage, and the soil is

**Table 1. Daily average evapotranspiration in the study area.**

| Date | Min. (mm/d) | Max. (mm/d) | Daily average ET (mm/d) | SD |
|---|---|---|---|---|
| 2017-01-02 | 0.02 | 3.44 | 0.56 | 0.3477 |
| 2016-04-21 | 0.15 | 12.58 | 2.02 | 1.4135 |
| 2017-05-26 | 0.35 | 15.13 | 3.51 | 2.1273 |
| 2017-07-13 | 0.51 | 14.56 | 5.17 | 2.0476 |
| 2016-09-28 | 0.22 | 8.07 | 3.31 | 1.1839 |
| 2016-12-17 | 0.07 | 3.36 | 0.72 | 0.3363 |

ET, evapotranspiration; Min., minimum; Max, maximum; SD, standard deviation.

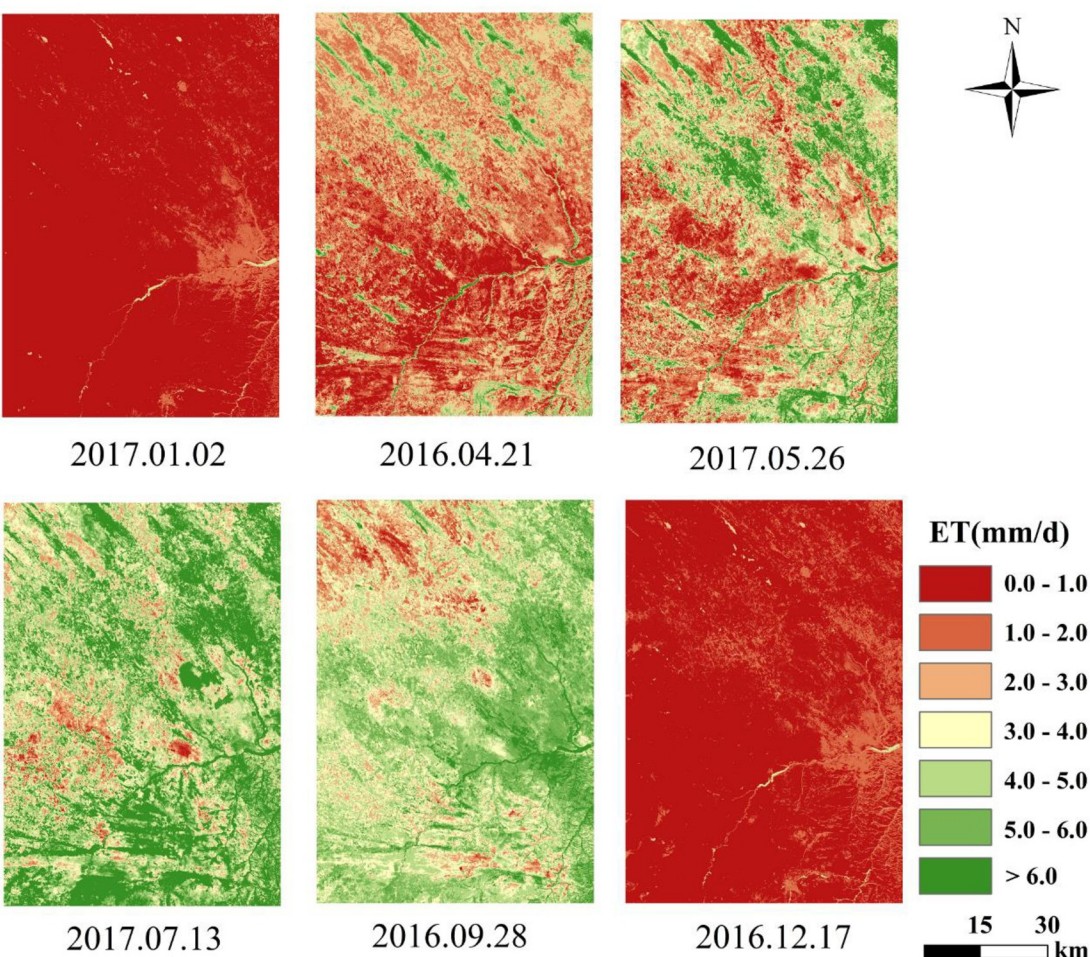

**Fig 3. Spatiotemporal distribution of daily evapotranspiration in the agro-pastoral ecotone of northwest China.**

dry and lacks water. From a local perspective, the daily ET of different land use types differs significantly.

## Discussion

### A correlation analysis of ET and meteorological factors

In this study, the annual daily data of the Hengshan station in 2016 were selected, including four meteorological factors: ET (mm/day), air temperature (TEM, ˚C), relative humidity (RHU, %), and wind speed (WIN, m/s). A correlation analysis between meteorological factors and the surface ET was performed using SPSS, and the strength of correlation between the meteorological factors and ET was evaluated using the correlation coefficient $R^2$. Fig 4 shows the linear relationship between ET and the meteorological factors. ET significantly correlated with the air temperature with an $R^2$ of 0.6274. The wind speed weakly correlated with an $R^2$ of 0.3307. The RHU had a negative $R^2$ of 0.3374. In areas with higher air humidity, the surface moisture faces more resistance when entering the air, so the ET was relatively small.

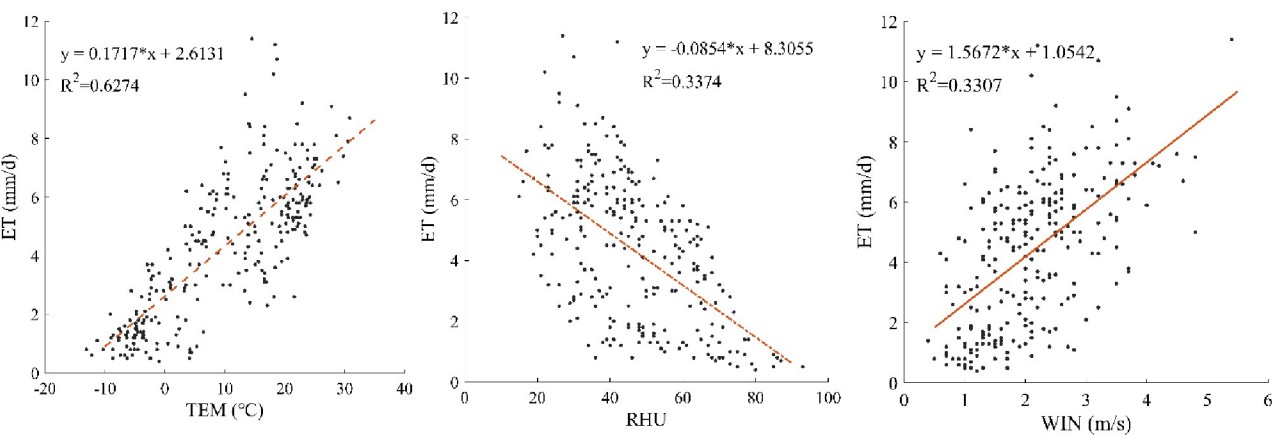

**Fig 4. The linear relationship between evapotranspiration and meteorological factors.**

## Sensitivity analysis of model parameters

To better understand the extent to which each model parameter affects the model inversion results, this study performed a sensitivity analysis of the model parameters to derive the amount of change in the final ET owing to changes in each parameter. The sensitivity analysis will be performed using the one-parameter variable method, i.e., the interaction of the parameters will not be considered. Only the parameters to be analyzed will be treated for variation. Since hydrothermal conditions have a large influence on ET, this study explored the sensitivity of atmospheric water vapor (wv) content, hot spot surface temperature ($LST_h$) and cold pixels to ET. The sensitivity index I was used to reflect the sensitivity of the evapotranspiration model to water vapor content. Larger values of I indicated that the model is highly sensitivity to this parameter. The formula is calculated as follows [32]:

$$I = \frac{\Delta ET}{\Delta X} \cdot \frac{X}{ET} \tag{13}$$

Where $\Delta X$ represents the amount of change in the parameter, and $\Delta ET$ represents the amount of change in the model output results owing to the change in the parameter. The sensitivity indices were classified into four classes by the magnitude of the value of I as shown in Table 2.

The inversion accuracy of wv and $LST_h$ were considered to be approximately 0.5 g/cm$^2$ and 2°C, respectively. $\Delta wv$ = 0.5 g/cm$^2$ and $\Delta LST$ = 2°C were established. Tables 3 and 4 show the mean values of the model outputs, as well as the sensitivity indices.

Table 3 shows the daily changes in ET and the sensitivity indices owing to changes in the wv. Increasing the wv by 0.5 g/cm$^2$ caused a daily change of 1.13 mm/d in the ET, and a decrease of 0.5 g/cm$^2$ wv caused a daily change of 0.88 mm/d in the ET. The sensitivity indices for increasing and decreasing the wv by $\Delta$, $2\Delta$, and $3\Delta$ were all > 0.2, indicating that the output

**Table 2. Sensitivity classification.**

| Grade | Sensitivity index | Sensitivity classification |
|-------|-------------------|----------------------------|
| I | 0.0<\|I\|<0.05 | poor |
| II | 0.05<\|I\|<0.20 | fair |
| III | 0.20<\|I\|<1.00 | moderate |
| IV | \|I\|>1.00 | excellent |

**Table 3. Evapotranspiration changes and sensitivity indices from the changes in water vapor.**

| Daily ET<br>wv | Average value (mm/d) | Average difference (mm/d) | \|I\| | Sensitivity classification |
|---|---|---|---|---|
| wv+1.5 g/cm$^2$ | 2.73 | -2.44 | 0.83 | moderate |
| wv+1.0 g/cm$^2$ | 3.24 | -1.93 | 0.98 | moderate |
| wv+0.5 g/cm$^2$ | 4.04 | -1.13 | 1.15 | excellent |
| wv-0.5 g/cm$^2$ | 6.05 | 0.88 | 0.90 | moderate |
| wv-1.0 g/cm$^2$ | 6.73 | 1.56 | 0.79 | moderate |
| wv-1.5g/cm$^2$ | 7.24 | 2.07 | 0.70 | moderate |

ET, evapotranspiration; wv, atmospheric water vapor.

of the SEBAL model-based inversion of daily ET is sensitive to the wv. The main reason is that the magnitude of wv directly affects the atmospheric transmittance, which causes a change in the net radiation flux. The effect on daily ET is weakened by the large changes in wv, which is owing to the large number of parameters in the SEBAL model. The changes in wv affect other parameters, thus, weakening the effect on daily ET.

Table 4 shows the daily changes in the ET and sensitivity indices owing to changes in the $LST_h$. A 2˚C increase in $LST_h$ caused a daily change of 0.15 mm/d in the ET. A 2˚C decrease in $LST_h$ caused an ET change of 0.08 mm/d. The sensitivity indices for increasing and decreasing the $LST_h$ by $\Delta$, $2\Delta$, and $3\Delta$ were all > 0.2, indicating that the inversion results are sensitive to the $LST_h$ based on the SEBAL model inversion of daily ET. The model output results on the $LST_h$ are less sensitive compared with those of the wv. The main reason for this is that the $LST_h$ is used to calculate the sensible heat flux, and the result of the sensible heat flux requires calculations that are continuously iterative. Multiple iterations weaken the influence of the $LST_h$. Thus, the change of a single parameter of the $LST_h$ does not have a large influence on the daily ET.

## Correlation analysis of the daily ET and surface parameters

An analysis of the spatiotemporal changes of ET in the study area indicated that the underlying surface characteristics, such as hydrothermal conditions and land use types, are important factors that affect energy transmission, and their interactions in concert affect the spatial distribution pattern of the surface ET [22]. To analyze the correlation between surface parameters and ET in more detail, this study obtained the spatial distribution map of the NDVI, LST, surface albedo and net radiant flux (Fig 5) and their correlation with the daily ET (Fig 6). The figures

**Table 4. Changes in evapotranspiration and sensitivity indices from the changes in hot surface temperature.**

| Daily ET<br>$LST_h$ | Average value (mm/d) | Average difference (mm/d) | \|I\| | Sensitivity classification |
|---|---|---|---|---|
| $LST_h$+6˚C | 5.54 | 0.37 | 0.56 | moderate |
| $LST_h$+4˚C | 5.42 | 0.25 | 0.57 | moderate |
| $LST_h$+2˚C | 5.30 | 0.13 | 0.59 | moderate |
| $LST_h$-2˚C | 5.09 | -0.08 | 0.36 | moderate |
| $LST_h$-4˚C | 5.05 | -0.13 | 0.30 | moderate |
| $LST_h$-6˚C | 4.97 | -0.20 | 0.30 | moderate |

ET, evapotranspiration; $LST_h$, hot surface temperature.

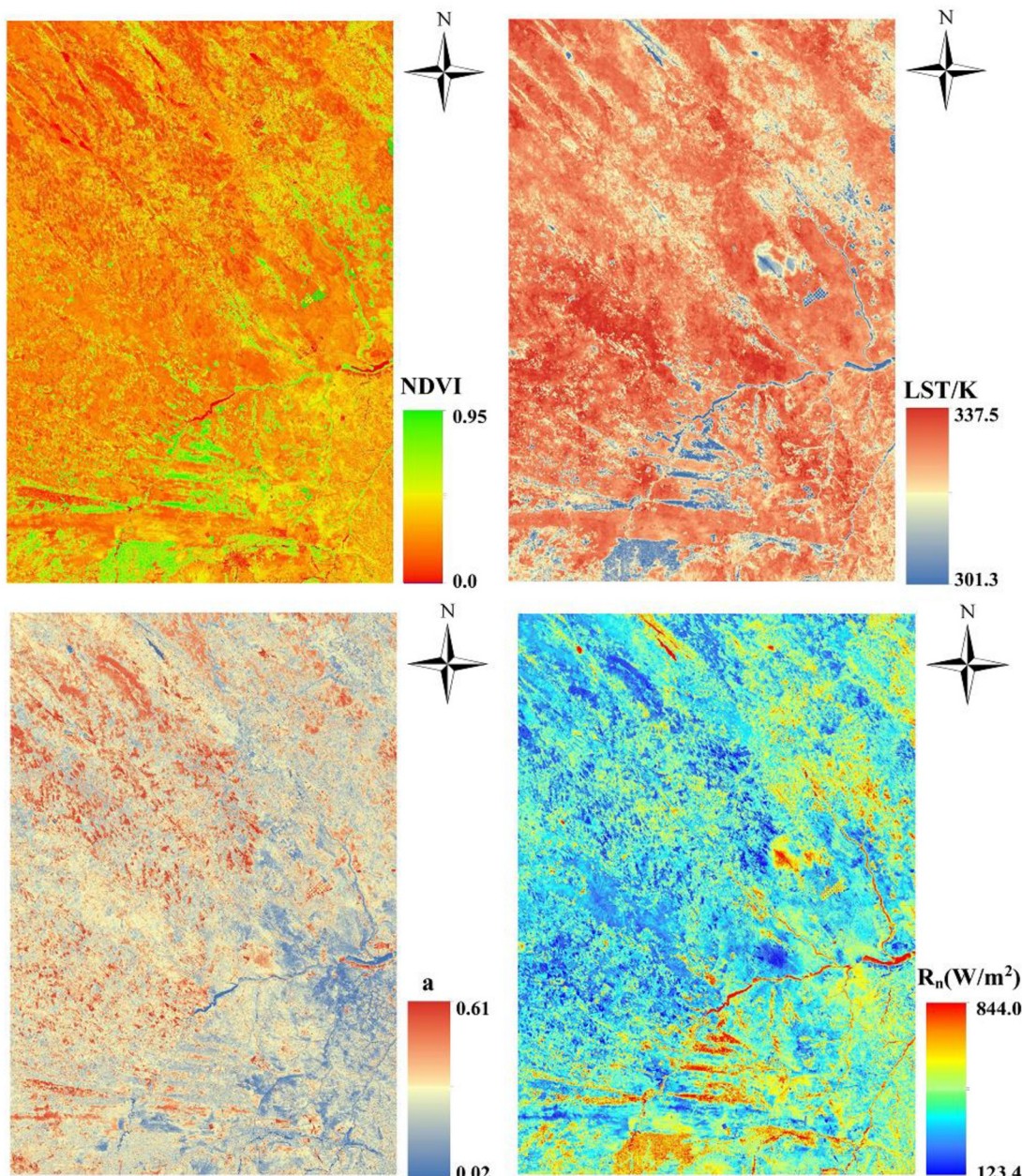

**Fig 5. Spatial distribution of the normalized vegetation index, surface temperature, surface albedo and the net radiation flux in the agro-pastoral ecotone of northwest China.**

indicate terms of spatial distribution. For example, the high-value areas of NDVI are primarily concentrated in the southern part of the study area and distributed in strips and points. This pattern is consistent with the spatial distribution of daily ET. The spatial distribution of surface temperature is opposite to that of the daily ET. The high values are distributed in the west and north of the study area on primarily bare and sandy lands. The high value areas of surface albedo are primarily distributed in the west and north, similar to the distribution of surface temperature. The main reason for this is that crops are grown in the south and east, and the

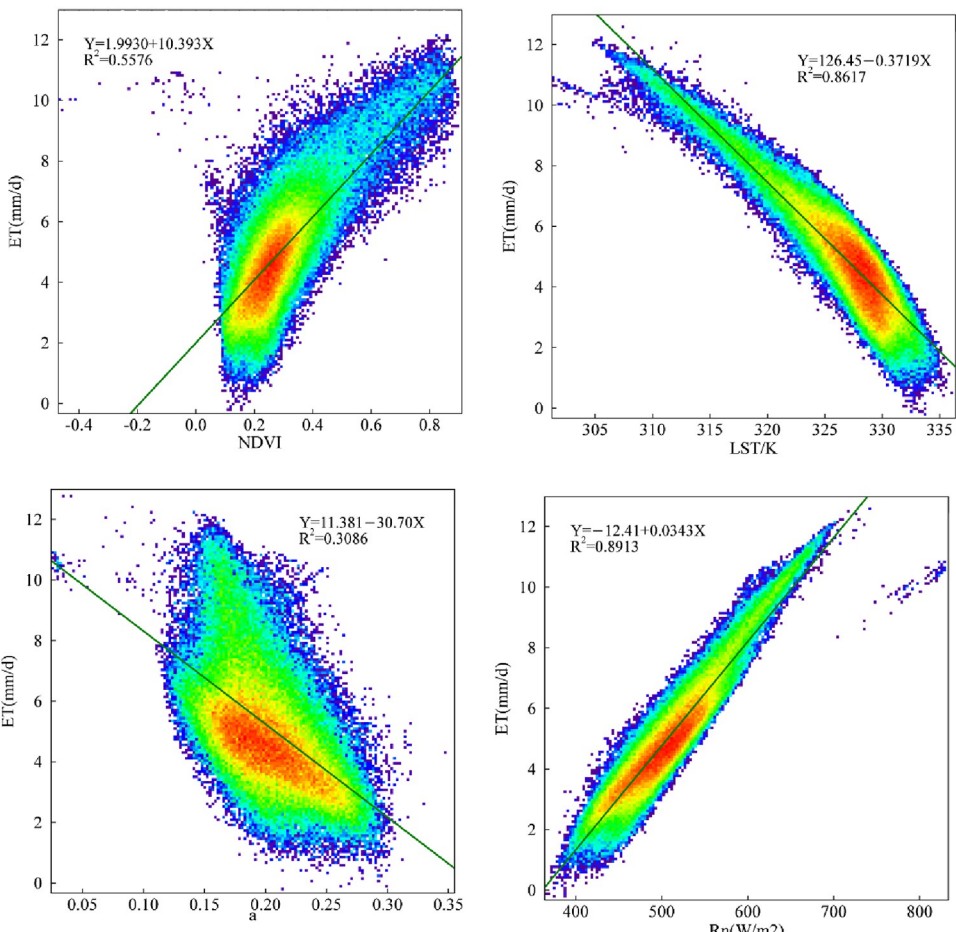

**Fig 6. Correlation between the daily evapotranspiration and surface parameters in the agro-pastoral ecotone of northwest China.**

northwest is primarily bare and sandy lands. The surface is not very rough, and there is strong reflection from solar energy. The pattern of spatial distribution of the net radiation flux and daily evaporation remains highly consistent. The correlation coefficients between the NDVI, LST, surface albedo, net radiant flux and daily ET were 0.5830, 0.8425, 0.3428, and 0.9111, respectively. Among them, the NDVI and net radiant flux positively correlated with the daily ET, and the LST and surface albedo negatively correlated with the daily ET. Overall, the correlation between daily ET and the underlying surface parameters from strong to weak is the net surface radiation > LST > NDVI > surface albedo. The correlation of surface parameters and daily ET as shown in this study has also been widely reported from many studies [6,33,34].

## Analysis of the daily ET with different land use types

Fig 7 shows the spatial distribution of land use types and daily ET in the agro-pastoral ecotone of northwest China. The amount of evapotranspiration varies from one land use type to another owing to differences in the physicochemical properties of the subsurface. The spatial distribution of the larger areas of ET was significantly similar to that of water and cropland. The spatial distribution of the smaller areas matches that of the bare land, and the daily ET of the built-up land did not differ significantly from that of the other surrounding land use types.

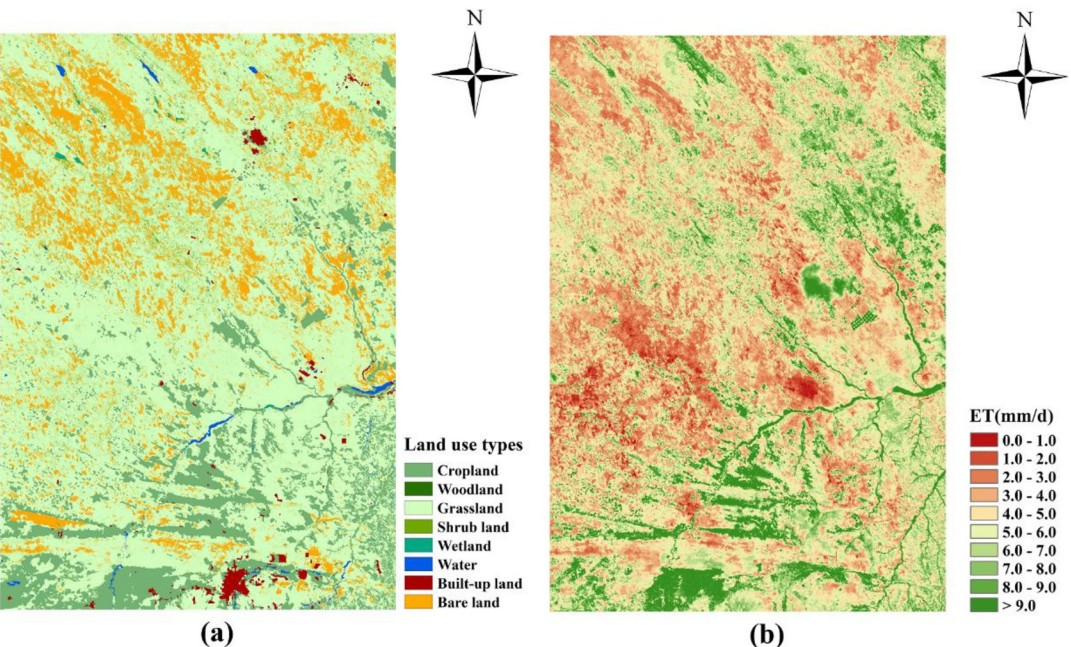

**Fig 7. Spatial distribution of (a) land use types and (b) daily evapotranspiration in agro-pastoral ecotone of northwest China.**

Table 5 shows the data of daily ET values for different land use types. The daily ET of water and woodland were the highest in study area, with average daily ET values of 9.90 mm/d and 8.69 mm/d, respectively, which is because water provides sufficient water resources. Woodland have a dual role of water connotation and ET and can provide good moisture conditions for ET, which should be relatively high. This result is consistent with a study by Jeremiah et al. [35]. The average daily ET of wetlands and croplands was 7.70 mm/d and 7.59 mm/d, respectively, which are relatively high categories of evapotranspiration. The combination of plant emissions in the results and evaporation from the soil and water surfaces results in high evapotranspiration. The croplands are predominantly dryland, and crop irrigation provides sufficient water for ET during warmer temperatures. The daily ET of built-up land, shrub land, grassland, and bare land were 6.39 mm/d, 5.74 mm/d, 4.92 mm/d and 3.78 mm/d, respectively.

**Table 5. Daily evapotranspiration data for different land use types.**

| Land use types | Area/km$^2$ | Percentage of total area/% | Daily average ET (mm/d) |
|---|---|---|---|
| Cropland | 1982.86 | 13.92 | 7.59 |
| Woodland | 9.40 | 0.07 | 8.69 |
| Grassland | 9571.70 | 67.21 | 4.92 |
| Shrub land | 312.56 | 2.19 | 5.74 |
| Wetland | 14.88 | 0.10 | 7.70 |
| Water | 55.89 | 0.39 | 9.90 |
| Built-up land | 126.57 | 0.89 | 6.39 |
| Bare land | 2167.58 | 15.22 | 3.78 |

ET, evapotranspiration.

Among them, shrub lands are dominated by plantation woodland that are young and have a low density and few species of trees. Therefore, its ET is relatively low. Grassland and bare land comprise approximately 67.21% of the study area, which is the dominant land use type in the study area. The grasslands are primarily desert and sandy vegetation, which are sparsely distributed. Owing to the unique mosaic substratum characteristics of the region, grassland and bare lands are distributed in intervals. Transpiration is low. Therefore, it has a small value for its daily ET. There is only a small area of built-up land, and the surrounding land use types are mostly cropland. As a result, the daily ET data for the built-up land are higher overall than the daily ET of grassland and shrub land. In summary, water and woodland are the two land use types that had the highest daily ET, indicating that water resources and vegetation cover are the main factors that determined the daily ET in the study area. Wetlands have a higher ET owing to the dual effect of plant ET and evaporation from soil and water surfaces, and cropland has a higher ET owing to irrigation and other integrated human management measures. Shrub land, grassland, and bare land have a lower daily ET owing to the sparse vegetation and insufficient soil moisture. The daily ET of built-up land is clearly overestimated. The exact reasons for this merit further analysis and research.

The verification of the ET model is an important part of the remote sensing inversion of ET. The use of remote sensing to estimate the surface flux in areas $< 1$ km$^2$ may have major problems, while the remote sensing inversion of large-scale areas is problematic owing to the problems of inadequate field observation data and errors in the observation instrument itself [14]. To more effectively vary the accuracy of the ET model, this study used the data of ET from the water surface of the weather station for relative verification and considered the problem of inconsistent water surface evaporators. The conversion coefficients of the E-601 type and 20cm-caliber small evaporator in the study area were explored using multiple sets of meteorological data, and the errors caused by the inconsistent evaporators were reduced. The conversion coefficients that were obtained are basically consistent with those in China provided by Mao Yue [36]. When the SEBAL model was used to retrieve the ET, the absolute error $< 1$ mm/d, indicating that the SEBAL model is highly applicable to retrieve the ET in agro-pastoral ecotone of northwest China. Similar results can also be found in other studies of different areas [6,37]. There is no doubt that the improvement in inversion accuracy of the SEBAL model merits celebration. However, surface parameters are important ones that affect the accuracy of the ET model. Norman et al. delineated the error possible from the ground temperature, i.e., with a speed of 5 m/s for a canopy that is 10 m high and an error of the ground temperature difference that exceeds 1 K, the error of H can reach 87 W/m$^2$ [38]. A study by Li et al. [22] found that the parameters with the strongest correlation between ET and surface characteristic parameters were surface temperature and net radiant flux, which are consistent with the results of this study. The uncertainty of current radiation temperature inversion is usually 1~3 K [39]. While the remote sensing ET model is widely used, it is also necessary to realize that there is still substantial uncertainty in the inversion of the surface parameters in the remote sensing ET model. Therefore, the future research of remote sensing ET models should ensure that each step of the inversion result and each surface parameter can be verified, so that the inversion result is accurate. The analysis of daily ET of different land use types found that the daily ET of built-up land was higher than that of the grasslands and shrub land. This may be owing to two reasons. First, the drought and water shortage in the study area can be owing to the sparseness of grasslands and shrub lands and their lower daily ET. The built-up land is mostly surrounded by cropland, which results in inaccurate data on the daily ET of built-up land. Secondly, the SEBAL model overestimates the inversion of the ET of built-up land, which is consistent with the results of Jin et al. [40].

## Conclusions

The SEBAL model was used to retrieve the daily surface ET based on Landsat8 remote sensing images in the northwestern windy sand and grassland area. This data was combined with that of the water surface evaporation from a weather station to verify the accuracy of the inversion. In addition, the correlation between the ET and surface parameters was analyzed to clarify the impact factor of surface ET. This study produced the following conclusions:

1. An analysis of the monitoring data of the E-601 type and 20cm-caliber small evaporator in the same period indicated that the average conversion coefficient of the two surface evaporators was 0.63, and the correlation coefficient $R^2$ was 0.95, indicating that the conversion coefficients of E-601 type and the 20cm-caliber small evaporator are highly reliable and can meet the conversion needs of the monitoring data of the two surface evaporators.

2. The actual surface ET retrieved based on the SEBAL model correlated highly with the surface water ET. The correlation coefficient was 0.81, and the average root mean square error was 0.9 mm/d, indicating that the SEBAL model that is used to retrieve surface ET is highly accurate in the agro-pastoral ecotone of northwest China.

3. The correlation coefficients between NDVI, LST, surface albedo and net radiant flux and daily ET were 0.5830, 0.8425, 0.3428, and 0.9111, respectively. Among them, the NDVI and net radiant flux positively correlated with the daily ET. The LST and surface albedo negatively correlated with the daily ET. The overall correlation between the daily ET and underlying surface parameters from strong to weak were the net radiant flux > LST > NDVI > surface albedo.

4. The amount of daily ET varied substantially in terms of spatial distribution. The daily ET of water was the highest, followed by woodland, wetland, cropland, built-up land, and shrubland. The area of grassland and bare land were the dominant land use types in the study area, but they had the lowest value of daily ET. The inversion of daily ET of built-up land based on the SEBAL model was overestimated.

## Supporting information

**S1 Dataset. Shape files of image coverage of four meteorological station.**
(ZIP)

**S2 Dataset. ET inversion image based on SEBAL model.**
(ZIP)

**S3 Dataset. Land use types in the study area.**
(ZIP)

## Acknowledgments

The data set is provided by Geospatial Data Cloud site, Computer Network Information Center, Chinese Academy of Sciences (http://www.gscloud.cn). And the authors gratefully acknowledge researchers at the Institute of Land Engineering and Technology, Shaan-xi Provincial Land Engineering Construction Group, for their help with the field experiments. We wish to thank the editor of this journal and the anonymous reviewers during the revision process.

## Author Contributions

**Conceptualization:** Liangyan Yang, Zenghui Sun.

**Data curation:** Yuanyuan Yang, Tong Li.

**Funding acquisition:** Zenghui Sun.

**Investigation:** Jianfeng Li.

**Methodology:** Liangyan Yang.

**Project administration:** Jinbao Liu.

**Resources:** Yuanyuan Yang, Tong Li.

**Software:** Liangyan Yang, Jianfeng Li.

**Supervision:** Jinbao Liu.

**Validation:** Liangyan Yang.

**Writing – original draft:** Liangyan Yang.

**Writing – review & editing:** Zenghui Sun, Jinbao Liu.

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
