## [Decision Letter · Decision Letter 0]

29 Oct 2021

PONE-D-21-26971Daily actual evapotranspiration estimation of different land use types based on SEBAL model in the agro-pastoral ecotone of northwest ChinaPLOS ONE

Dear Dr. Yang,

Thank you for submitting your manuscript to PLOS ONE. After careful consideration, we feel that it has merit but does not fully meet PLOS ONE’s publication criteria as it currently stands. Therefore, we invite you to submit a revised version of the manuscript that addresses the points raised during the review process.

ACADEMIC EDITOR:

We have received comments from two reviewers on your manuscript. 

The reviewers favor accepting your paper, although both propose quite a few technical corrections to your text. Please, include them in your paper. In your responses to the reviewers, please reply individually and address their remarks point by point. Make sure that you either explicitly indicate the action taken following a particular reviewer’s comment or clearly explain your disagreement with the reviewer.

We look forward to receiving your revised manuscript.

Kind regards,

Débora Regina Roberti, Ph.D.

Academic Editor

PLOS ONE

Journal Requirements:

This research was funded by Technology Innovation Center for Land Engineering and Human Settlements, Shaanxi Land Engineering Construction Group Co., Ltd and Xian Jiaotong University (2021WHZ0088, 2021WHZ0091) and Scientific Research Item of Shaanxi Provincial Land Engineering Built-up Group (DJNY2021-33).

3. Thank you for stating the following in the Acknowledgments / Funding Section of your manuscript: 

This research was funded by Technology Innovation Center for Land Engineering and Human Settlements, Shaanxi Land Engineering Construction Group Co., Ltd and Xian Jiaotong University (2021WHZ0088, 2021WHZ0091) and Scientific Research Item of Shaanxi Provincial Land Engineering Built-up Group (DJNY2021-33).

This research was funded by Technology Innovation Center for Land Engineering and Human Settlements, Shaanxi Land Engineering Construction Group Co., Ltd and Xian Jiaotong University (2021WHZ0088, 2021WHZ0091) and Scientific Research Item of Shaanxi Provincial Land Engineering Built-up Group (DJNY2021-33).

4. We note that Figure 1, 3, 4 and 6 in your submission contain [map/satellite] images which may be copyrighted. All PLOS content is published under the Creative Commons Attribution License (CC BY 4.0), which means that the manuscript, images, and Supporting Information files will be freely available online, and any third party is permitted to access, download, copy, distribute, and use these materials in any way, even commercially, with proper attribution. For these reasons, we cannot publish previously copyrighted maps or satellite images created using proprietary data, such as Google software (Google Maps, Street View, and Earth). For more information, see our copyright guidelines: http://journals.plos.org/plosone/s/licenses-and-copyright.

a. You may seek permission from the original copyright holder of Figure 1, 3, 4 and 6 to publish the content specifically under the CC BY 4.0 license.  

Reviewers' comments:

Reviewer's Responses to Questions

**Comments to the Author**

1. Is the manuscript technically sound, and do the data support the conclusions?

Reviewer #1: Partly

Reviewer #2: Yes

2. Has the statistical analysis been performed appropriately and rigorously? 

Reviewer #1: No

Reviewer #2: Yes

3. Have the authors made all data underlying the findings in their manuscript fully available?

Reviewer #1: No

Reviewer #2: Yes

4. Is the manuscript presented in an intelligible fashion and written in standard English?

Reviewer #1: No

Reviewer #2: No

5. Review Comments to the Author

Reviewer #1: This paper was used SEBAL model to estimate ET by Landsat imagery. The topic is quite interesting and could be stimulating for the scientific community after the major revision.

1. As a general comment, I can say that the paper/work is still in an initial phase of implementation, and deeper analysis is needs.

2. The novelty, originality, and scientific contribution of this study in its respective field is not well presented. I think authors should clarify why the study is important, i.e., what is the contribution of this study inside of its branch?

3. Can Authors explain the verification section in more detail?

- How many ground stations were used for validation? (Line 172 mentioned 4. But 8 station were mentioned between line 180-181)

- What is temporal resolution of station data?

- What is error of the data verification? How the author prevents the errors of data validation?

4. E-601 and 20cm-caliber have proven “meet the requirements” (line 185), why only 20cm-caliber data was selected as the validation data?

5. Please please clarify the fig 2 in more detail. Author mentioned 4 stations, investigated 6 days, why were 12 “point” used in the competition?

6. The results section is not clear and need to be improved.

- The solid explanation should be given instead of rather generic explanations are provided (“Spatiotemporal distribution of daily ET” section).

- There are many surface parameters affect to ET, such as wind speed, air humidity…. Why four factor, including NDVI, LST, land surface albedo, net radiation flux, were selected?

- NDVI, LST, land surface albedo, net radiation flux are inputs the SEBAL model, so it's a problem of measuring similarities.

Reviewer #2: The paper is a straightforward analysis for invert the regional ET based on Landsat8 images and compare with observed ET and explore the mechanisms of surface parameters on daily ET. I recommend publication after the following minor revisions.

(1) The latent heat flux should be calculated by P-M equation

(2) The author should clarify the distinction between daily ET and latent heat flux

(3) It is interesting to explore the mechanism of how hydrothermal conditions effect the energy transmission.

(4) Please explain why the results of albedo v.s. ET shows a larges bias

6. PLOS authors have the option to publish the peer review history of their article (what does this mean?). If published, this will include your full peer review and any attached files.

Reviewer #1: No

Reviewer #2: No

---

## [Author Response · Author response to Decision Letter 0]

8 Dec 2021

Response to reviewers’ comments

We would like to thank the academic editor and reviewers for their constructive and insightful comments and suggestions. We have revised the manuscript accordingly. To facilitate the reviewers to see the changes we made in response to their comments, the changes are marked in blue in the revised manuscript. Below we provide a point to point response to all the comments.

Academic Editor

Question 1: Please ensure that your manuscript meets PLOS ONE's style requirements, including those for file naming. The PLOS ONE style templates can be found at https://journals.plos.org/plosone/s/file?id=wjVg/PLOSOne_formatting_sample_main_body.pdf and https://journals.plos.org/plosone/s/file?id=ba62/PLOSOne_formatting_ sample_title_authors_affiliations.pdf

Response: Thanks for the suggestion. We have revised the whole article according to PLOS ONE style templates. 

Question 2: Thank you for stating the following financial disclosure:

This research was funded by Technology Innovation Center for Land Engineering and Human Settlements, Shaanxi Land Engineering Construction Group Co., Ltd and Xian Jiaotong University (2021WHZ0088, 2021WHZ0091) and Scientific Research Item of Shaanxi Provincial Land Engineering Built-up Group (DJNY2021-33).

Response: Thanks for the suggestion. We have updated the Funding Statement and Competing Interests Statement in cover letter.

Funding: This paper was supported by the Fund for Less Developed Regions of the National Natural Science Foundation of China (No. 42167039), Technology Innovation Center for Land Engineering and Human Settlements, Shaanxi Land Engineering Construction Group Co., Ltd and Xian Jiaotong University (2021WHZ0088, 2021WHZ0091) and Scientific Research Item of Shaanxi Provincial Land Engineering Built-up Group (DJNY2021-33). The funder provided support in the form of salaries for authors [Yang LY, Li JF, Sun ZH, Liu JB], but did not have any additional role in the study design, data collection and analysis, decision to publish, or preparation of the manuscript. The specific roles of these authors are articulated in the ‘author contributions’ section.

Competing interests: Institute of Land Engineering and Technology, Shaanxi Provincial Land Engineering Construction Group Co., Ltd and Shaanxi Provincial Land Engineering Construction Group Co., Ltd do not alter our adherence to PLOS ONE policies on sharing data and materials. The authors have declared that no competing interests exist.

Question 3: Thank you for stating the following in the Acknowledgments / Funding Section of your manuscript: 

This research was funded by Technology Innovation Center for Land Engineering and Human Settlements, Shaanxi Land Engineering Construction Group Co., Ltd and Xian Jiaotong University (2021WHZ0088, 2021WHZ0091) and Scientific Research Item of Shaanxi Provincial Land Engineering Built-up Group (DJNY2021-33).

This research was funded by Technology Innovation Center for Land Engineering and Human Settlements, Shaanxi Land Engineering Construction Group Co., Ltd and Xian Jiaotong University (2021WHZ0088, 2021WHZ0091) and Scientific Research Item of Shaanxi Provincial Land Engineering Built-up Group (DJNY2021-33).

Response: Thanks for the suggestion. We have updated the Funding Statement and Competing Interests Statement in cover letter. And remove any funding-related text from the manuscript.

Funding: This paper was supported by the Fund for Less Developed Regions of the National Natural Science Foundation of China (No. 42167039), Technology Innovation Center for Land Engineering and Human Settlements, Shaanxi Land Engineering Construction Group Co., Ltd and Xian Jiaotong University (2021WHZ0088, 2021WHZ0091) and Scientific Research Item of Shaanxi Provincial Land Engineering Built-up Group (DJNY2021-33). The funder provided support in the form of salaries for authors [Yang LY, Li JF, Sun ZH, Liu JB], but did not have any additional role in the study design, data collection and analysis, decision to publish, or preparation of the manuscript. The specific roles of these authors are articulated in the ‘author contributions’ section.

Competing interests: Institute of Land Engineering and Technology, Shaanxi Provincial Land Engineering Construction Group Co., Ltd and Shaanxi Provincial Land Engineering Construction Group Co., Ltd do not alter our adherence to PLOS ONE policies on sharing data and materials. The authors have declared that no competing interests exist.

Question 4: We note that Figure 1, 3, 4 and 6 in your submission contain [map/satellite] images which may be copyrighted. All PLOS content is published under the Creative Commons Attribution License (CC BY 4.0), which means that the manuscript, images, and Supporting Information files will be freely available online, and any third party is permitted to access, download, copy, distribute, and use these materials in any way, even commercially, with proper attribution. For these reasons, we cannot publish previously copyrighted maps or satellite images created using proprietary data, such as Google software (Google Maps, Street View, and Earth). For more information, see our copyright guidelines: http://journals.plos.org/plosone/s/licenses-and-copyright.

Response: Thanks for the suggestion. The remote sensing images used in this study are all freely available at the Geospatial Data Cloud (http://www.gscloud.cn). We have explained the acquisition of Data sources of the revised manuscript.

Reviewer #1 This paper was used SEBAL model to estimate ET by Landsat imagery. The topic is quite interesting and could be stimulating for the scientific community after the major revision.

Questions 1: As a general comment, I can say that the paper/work is still in an initial phase of implementation, and deeper analysis is needs.

Response: We are extremely grateful to reviewer for pointing out this problem. To be more clearly and in accordance with the reviewer concerns, we have added a more detailed discussion of meteorological factors and hydrothermal conditions. In future research, we will deeply analyze the mechanism of remote sensing inversion of evapotranspiration and improve the accuracy of model inversion. It can provide data support for the rational use of water resources, restoration of the ecological environment, and protection of ecological water sources. Thank you again for your support and affirmation of our work.

Questions 2: The novelty, originality, and scientific contribution of this study in its respective field is not well presented. I think authors should clarify why the study is important, i.e., what is the contribution of this study inside of its branch? 

Response: Evapotranspiration is an important part of the water cycle. Remote sensing technology provides a new method for the estimation of regional land surface water evapotranspiration. This study uses Landsat data and energy balance models to estimate the evapotranspiration in the agro-pastoral ecotone, which solves the problem of transient remote sensing to estimate all-day evapotranspiration. Through the simulation of the daily evapotranspiration in the study area, the distribution law of evapotranspiration in this area is analyzed, which has certain support and guiding significance for the study of land surface process. In the context of increasingly scarce global water resources, accurate estimation of evapotranspiration is not only of great significance for theoretical research on global climate evolution, environmental issues, and evaluation of water resources, but also for guiding agricultural drainage and irrigation, monitoring agricultural drought, and improving agricultural utilization rate of water resources is also very practical significance.

Questions 3: Can Authors explain the verification section in more detail?

- How many ground stations were used for validation? (Line 172 mentioned 4. But 8 station were mentioned between line 180-181)

- What is temporal resolution of station data?

- What is error of the data verification? How the author prevents the errors of data validation?

Response: Our deepest gratitude goes to you for your careful work and thoughtful suggestions that have helped improve this paper substantially. There are four weather stations in the study area, plus 4 surrounding weather stations, a total of eight stations. Since the meteorological sites were small meteorological stations that were established by research teams in Wushenqi and Henan in China, they did not have evaporation dishes to obtain the data on water surface evaporation. Therefore, E-601 and 20cm-caliber meteorological stations, such as those at the Hengshan, Jingbian, Shenmu, Dingbian, Yanchi and Etuoke Banner Stations in China were used for validation in the study and surrounding areas. The water surface evaporator of the weather station has two types that include E-601 and 20cm-caliber. There are differences in the installation plan, structure and observational methods of the two instruments. The two monitoring results are quite different and cannot be directly used to verify the accuracy of the evapotranspiration inversion results. Therefore, to more accurately explore the inversion accuracy of the SEBAL model and maintain the continuity and consistency of the data at each station, it is necessary to explore the conversion coefficients of the two surface evaporators and unify the data. So as to avoid the limitations brought by the data itself.

Questions 4: E-601 and 20cm-caliber have proven “meet the requirements” (line 185), why only 20cm-caliber data was selected as the validation data?

Response: We are very sorry for the inconvenience and confusion caused by my presentation. At present, most weather stations in China have two types of evaporating pans, E-601 and 20cm-caliber, which leads to inconsistent evapotranspiration data. In order to meet the needs, I explored the conversion coefficients of the two by using the weather stations around the study area. Data unification. And because the E-601 evaporating pan monitoring data is closer to the water surface evapotranspiration, this study converted the 20cm-caliber data into E-601 data, and used E-601 data as the verification data. In order to avoid unnecessary mistakes, I have sought out English-speaking experts with professional backgrounds to polish the language.

Questions 5: Please please clarify the fig 2 in more detail. Author mentioned 4 stations, investigated 6 days, why were 12 “point” used in the competition?

Response: Since the meteorological sites were small meteorological stations that were established by research teams in Wushenqi and Henan in China, they did not have evaporation dishes to obtain the data on water surface evaporation. The data involved in the validation were the monitoring data from the meteorological stations with a total of 12 images in 6 scenes in Hengshan and Jingbian, China. We have added explanations of accuracy verification of daily ET in Section 3.1 of the revised manuscript.

Questions 6: The results section is not clear and need to be improved.

- The solid explanation should be given instead of rather generic explanations are provided (“Spatiotemporal distribution of daily ET” section).

- There are many surface parameters affect to ET, such as wind speed, air humidity…. Why four factor, including NDVI, LST, land surface albedo, net radiation flux, were selected?

- NDVI, LST, land surface albedo, net radiation flux are inputs the SEBAL model, so it's a problem of measuring similarities

Response: The suggestion is taken. Our deepest gratitude goes to you for your careful work and thoughtful suggestions that have helped improve this paper substantially. We re-organized the content of the “Spatiotemporal distribution of daily ET” section, and explained the seasonal variation process and spatial distribution of daily evapotranspiration. In summary, hydrothermal conditions are the main factor that affect the changes in evapotranspiration during the year. From a local perspective, the daily ET of different land use types differs significantly. The characteristics of the underlying surface are important factors that affect the exchange of energy and material on the surface. NDVI, surface temperature, surface albedo and net surface radiation are several important parameters describing the nature of the underlying surface. They interact and jointly affect the spatial distribution pattern of evapotranspiration. Therefore, I chose NDVI, LST, land surface albedo and net radiation flux to explore the influencing factors of evapotranspiration. In order to better clarify the factors affecting evapotranspiration, we added Sensitivity analysis of model parameters, including hot spot surface temperature and water vapor. We have explained the details in Sections 4.2 and 4.3 of the revised manuscript.

Reviewer #2: The paper is a straightforward analysis for invert the regional ET based on Landsat8 images and compare with observed ET and explore the mechanisms of surface parameters on daily ET. I recommend publication after the following minor revisions.

Questions 1: The latent heat flux should be calculated by P-M equation

Response: The P-M formula is a standard formula for calculating the reference crop water requirement. The P-M can be used to calculate the reference crop evapotranspiration at a single point, and sufficient meteorological parameters are required as support. The latent heat flux obtained in this study is regional, and the meteorological data is not abundant, so the SEBAL model calculation is more suitable. 

Questions 2: The author should clarify the distinction between daily ET and latent heat flux.

Response: Latent Heat Flux is the heat exchange per unit area under the condition of constant temperature, in watts per square meter (W/m2); daily ET is the amount of water actually evaporated in a day (mm/d), which is the result of dividing the total latent heat flux of the whole day by the latent heat of vaporization (λ) of water. We have added explanations of the latent heat flux and daily ET in Section 2.3.4 and2.3.5 of the revised manuscript.

Questions 3: It is interesting to explore the mechanism of how hydrothermal conditions effect the energy transmission.

Response: Thanks for the suggestion. The suggestion is taken. This research has added Sensitivity analysis of model parameters in the discussion section. The influence of hotspot surface temperature and water vapor content on the model results is discussed, and the importance of hydrothermal conditions to SEBAL is further proved. The results show that the model output results on the LSTh are less sensitive compared with those of the wv. The main reason for this is that the LSTh is used to calculate the sensible heat flux, and the result of the sensible heat flux requires calculations that are continuously iterative. Multiple iterations weaken the influence of the LSTh. Thus, the change of a single parameter of the LSTh does not have a large influence on the daily ET. See the discussion section in the text for detailed analysis results.

Questions 4: Please explain why the results of albedo v.s. ET shows a larges bias 

Response: The surface albedo determines the amount of solar energy received at the pixel scale. The higher the surface albedo, the smaller the effective radiation reaching the surface, the less energy used for evapotranspiration, and the smaller the evapotranspiration. Therefore, there is a negative correlation between surface albedo and evapotranspiration, which is consistent with the results reflected in Figure 4. However, ET is affected by both energy and moisture. In arid areas, moisture is the decisive factor limiting the amount of surface evapotranspiration. Water in the study area is not sufficient, which limits the amount of evapotranspiration, which results in a large deviation between the surface albedo and ET.

---

## [Decision Letter · Decision Letter 1]

9 Feb 2022

PONE-D-21-26971R1Daily actual evapotranspiration estimation of different land use types based on SEBAL model in the agro-pastoral ecotone of northwest ChinaPLOS ONE

Dear Dr. Yang,

Thank you for submitting your manuscript to PLOS ONE. After careful consideration, we feel that it has merit but does not fully meet PLOS ONE’s publication criteria as it currently stands. Therefore, we invite you to submit a revised version of the manuscript that addresses the points raised during the review process.

Please submit your revised manuscript by  February 22, 2022 . If you will need more time than this to complete your revisions, please reply to this message or contact the journal office at plosone@plos.org. Please include the following items when submitting your revised manuscript:A rebuttal letter that responds to each point raised by the academic editor and reviewer(s). You should upload this letter as a separate file labeled 'Response to Reviewers'.A marked-up copy of your manuscript that highlights changes made to the original version. You should upload this as a separate file labeled 'Revised Manuscript with Track Changes'.An unmarked version of your revised paper without tracked changes. You should upload this as a separate file labeled 'Manuscript'.If applicable, we recommend that you deposit your laboratory protocols in protocols.io to enhance the reproducibility of your results. Protocols.io assigns your protocol its own identifier (DOI) so that it can be cited independently in the future. For instructions see: https://journals.plos.org/plosone/s/submission-guidelines#loc-laboratory-protocols. Additionally, PLOS ONE offers an option for publishing peer-reviewed Lab Protocol articles, which describe protocols hosted on protocols.io. Read more information on sharing protocols at https://plos.org/protocols?utm_medium=editorial-email&utm_source=authorletters&utm_campaign=protocols.

We look forward to receiving your revised manuscript.

Kind regards,

Débora Regina Roberti, Ph.D.

Academic Editor

PLOS ONE

Journal Requirements:

Reviewers' comments:

Reviewer's Responses to Questions

**Comments to the Author**

1. If the authors have adequately addressed your comments raised in a previous round of review and you feel that this manuscript is now acceptable for publication, you may indicate that here to bypass the “Comments to the Author” section, enter your conflict of interest statement in the “Confidential to Editor” section, and submit your "Accept" recommendation.

Reviewer #1: All comments have been addressed

2. Is the manuscript technically sound, and do the data support the conclusions?

Reviewer #1: Yes

3. Has the statistical analysis been performed appropriately and rigorously? 

Reviewer #1: Yes

4. Have the authors made all data underlying the findings in their manuscript fully available?

Reviewer #1: Yes

5. Is the manuscript presented in an intelligible fashion and written in standard English?

Reviewer #1: Yes

6. Review Comments to the Author

Reviewer #1: Dear authors, I do appreciate your further efforts in producing an advanced version of your paper that takes into account my comments.

The paper has improved. I suggest accepting it, after the authors make the following necessary revision:

- Normalized Difference Latent Heat (NDLI) index was proposed and proved as a reference for the assessment and monitoring water availability. you are encouraged to use it for your validation.

- Line 30 : “… coefficients of normalized vegetation indices…” Only NDVI was used for validation as spectral indices…so used as singular noun

- Please, improve the quality of the figure (as grip, scale…)

7. PLOS authors have the option to publish the peer review history of their article (what does this mean?). If published, this will include your full peer review and any attached files.

Reviewer #1: **Yes: **Mai Son Le

---

## [Author Response · Author response to Decision Letter 1]

13 Feb 2022

Response to reviewers’ comments

We would like to thank the academic editor and reviewers for their constructive and insightful comments and suggestions. We have revised the manuscript accordingly. To facilitate the reviewers to see the changes we made in response to their comments, the changes are marked in blue in the revised manuscript. Below we provide a point-to-point response to all the comments.

Academic Editor

Question 1: Please review your reference list to ensure that it is complete and correct. If you have cited papers that have been retracted, please include the rationale for doing so in the manuscript text, or remove these references and replace them with relevant current references. Any changes to the reference list should be mentioned in the rebuttal letter that accompanies your revised manuscript. If you need to cite a retracted article, indicate the article’s retracted status in the References list and also include a citation and full reference for the retraction notice.

Response: Our deepest gratitude goes to you for your careful work and thoughtful suggestions that have helped improve this paper substantially. We have revised the reference list to ensure that it is complete and correct. The changes of the reference list are marked in blue in the Revised Manuscript with Track Changes.

Reviewer：Dear authors, I do appreciate your further efforts in producing an advanced version of your paper that takes into account my comments. 

The paper has improved. I suggest accepting it, after the authors make the following necessary revision:

Questions 1: Normalized Difference Latent Heat (NDLI) index was proposed and proved as a reference for the assessment and monitoring water availability. you are encouraged to use it for your validation.

Response: Thank you again for your support and affirmation of our work. Thank you very much for your suggestion. Since we have not used the Normalized Difference Latent Heat (NDLI) index in past research, this study has not used it for verification. Thank you for pointing us in new directions. And in future research work, we will introduce the Normalized Difference Latent Heat (NDLI) index to improve our research results. Thank you again for your affirmation and support of this research.

Questions 2: Line 30 : “… coefficients of normalized vegetation indices…” Only NDVI was used for valiation as spectral indices…so used as singular noun.

Response: We are extremely grateful to reviewer for pointing out this problem. We have fixed this question. “… coefficients of normalized vegetation indices…” modified to “… coefficients of normalized vegetation index…”. Thank you again for your support and affirmation of our work.

Questions 3: Please, improve the quality of the figure (as grip, scale…).

Response: Thanks for the suggestion. We have improved the quality of Figures 1, 2, 4, and 6, where Figure 1 has modified the scale, and Figures 2 and 4 have increased the resolution of the figures.

---

## [Editor Report · Decision Letter 2]

24 Feb 2022

Daily actual evapotranspiration estimation of different land use types based on SEBAL model in the agro-pastoral ecotone of northwest China

PONE-D-21-26971R2

Dear Dr. Yang,

We’re pleased to inform you that your manuscript has been judged scientifically suitable for publication and will be formally accepted for publication once it meets all outstanding technical requirements.

Kind regards,

Débora Regina Roberti, Ph.D.

Academic Editor

PLOS ONE
---

## [Editor Report · Acceptance letter]

4 Mar 2022

PONE-D-21-26971R2 

Daily actual evapotranspiration estimation of different land use types based on SEBAL model in the agro-pastoral ecotone of northwest China 

Dear Dr. Yang:

I'm pleased to inform you that your manuscript has been deemed suitable for publication in PLOS ONE. Congratulations! Your manuscript is now with our production department. 

Kind regards, 

on behalf of

Dr. Débora Regina Roberti 

Academic Editor

PLOS ONE